# Effect of Fever on the Clinical Outcomes of Traumatic Brain Injury by Age

**DOI:** 10.3390/medicina58121860

**Published:** 2022-12-16

**Authors:** Dahae Lee, Hyunho Ryu, Eujene Jung

**Affiliations:** 1Chonnam National University Hospital, Gwangju 61186, Republic of Korea; 2Chonnam National University, Gwangju 61186, Republic of Korea

**Keywords:** traumatic brain injury, fever, outcome

## Abstract

*Background and objective*: Fever is a common symptom in patients with traumatic brain injury (TBI). However, the effect of fever on the clinical outcomes of patients with TBI is not well characterized. Our study aims to determine the impact of fever on the clinical outcomes of patients with TBI and test the interaction effect of fever on study outcomes according to age group. *Materials and methods*: Our retrospective study included adult patients with TBI who were transported to a level 1 trauma center by the emergency medical services (EMS) team. The main exposure is fever, defined as a body temperature of 38 °C or above, in the emergency department (ED). The primary outcome was mortality at hospital discharge. We conducted a multivariable logistic regression analysis to estimate the effect sizes of fever on study outcomes. We also conducted an interaction analysis between fever and age group on study outcomes. *Results*: In multivariable logistic regression analysis, patients with TBI who had fever showed no significant difference in mortality at hospital discharge (aOR, 95% CIs: 1.24 (0.57–3.02)). Fever significantly increased the mortality of elderly patients (>65 years) with TBI (1.39 (1.13–1.50)), whereas there was no significant effect on mortality in younger patients (18–64 years) (0.85 (0.51–1.54)). *Conclusions*: Fever was associated with mortality only in elderly patients with TBI.

## 1. Introduction

Traumatic brain injury (TBI) is a global burden and is the leading cause of death and disability worldwide [1]. The global incidence rate of TBI is estimated at more than 200 per 100,000 people each year. The general case fatality rate ranges from 0.9 to 7.6 per 100 patients with TBI worldwide [2]. 

The prediction of clinical outcomes of patients with TBI is difficult, to date. It has been reported that patients’ characteristics including age, Glasgow coma scale (GCS), pupil reflex, serum glucose level, computed tomographic (CT) classification, and serum or cerebrospinal fluid biomarker level may predict the clinical outcomes of patients with TBI [3,4].

Fever is a common symptom in TBI, which may occur in 20–50% of patients with TBI, and fever may occur due to various causes other than infection [5]. Previous studies have demonstrated that brain injury causes fever in the long term; however, few studies have investigated the effect of fever at the time of TBI on clinical outcomes. Although the relationship between fever at the time of TBI and the clinical outcomes of TBI is complex, fever has consistently been shown to be associated with an increased length of intensive care unit (ICU) and hospital stay and a higher mortality rate in previous studies. Every 1 °C increase in temperature has been associated with an increased risk of adverse outcomes by 2.2-fold. Further, a 0.5 °C rise in temperature can lead to a series of secondary injuries and neuron death [6]. These findings are consistent with those of the meta-analysis [7].

Poor outcomes may be due to elevated levels of excitatory amino acids including dopamine and glutamate, pyruvate, lactic acids, and free radicals; impaired enzymatic function; and blood–brain barrier breakdown on a local level, which may lead to cerebral edema, potentially decreasing cerebral perfusion pressure (CPP) [7,8].

Older adults are exposed to a higher incidence of TBI because of anatomical changes to the dura and more frequently receive aspirin and anticoagulant therapies than younger individuals [9]. Further, older age has been recognized as an independent predictor of poor outcomes of TBI [10]. Although the definite mechanism by which this occurs remains unknown, cerebral edema formation in brain-injured mice is reported to increase faster and become more severe with increasing age [11].

Based on the results of previous studies reporting that fever at the time of TBI causes cerebral edema to decrease CPP and that cerebral edema occurs more rapidly with increasing age [12,13,14], we hypothesized that fever burden is a predictor of poor outcome in patients with TBI, and this effect differs across age groups.

Our study aims to determine the impact of fever on the clinical outcomes of patients with TBI and test the interaction effect of fever on study outcomes according to age group.

## 2. Materials and Methods

### 2.1. Study Design, Setting, and Data Sources

The present investigation was a retrospective study of patients with TBI who were transported to Chonnam National University Hospital (CNUH) level 1 trauma center in South Korea by EMS. In Korea, the scoop-and-run system is used when encountering injured patients; after the measurement of vital signs, physical examination, and the provision of essential treatment such as a neck brace, simple dressing, or splint, the patient is transported to a hospital or a trauma center. If massive bleeding is suspected or hypotension is observed, fluid treatment is performed under the medical supervision of the emergency physician. If major trauma is suspected based on a comprehensive assessment of vital signs, mental state, and mechanism of injury, the patient is transported to a level 1 trauma center; if it is difficult to assess, the patient is provided medical control [15].

Since the CNUH established a trauma center in 2010, it has been operating with a team of specialists from diverse departments. If a patient is suspected of having sustained a TBI, it is reported to the hospital by paramedics at the scene or during transport, and, after arriving at the hospital, computed tomography (CT) of the brain is performed after an emergency medical physician administers emergency treatment. Neurosurgeons affiliated with the trauma center for brain hemorrhages and other concomitant injuries are notified simultaneously with general, thoracic, and orthopedic surgeons; if necessary, surgery is performed immediately. If blood transfusion or the use of drugs such as mannitol is required in the emergency department (ED), it is performed under the judgment of the emergency physicians. If fever is observed in the patient, fluids and antipyretics are administered. These protocols are maintained 24 h per day throughout the year.

The present study was approved by the Institutional Review Board of CNUH. Owing to the retrospective nature of the study and the use of anonymized patient data, the requirement for informed consent was waived.

### 2.2. Study Population

This study was conducted on patients with TBI over 18 years of age who visited a level 1 trauma center using emergency medical services (EMS) between January 2015 and December 2020. Patients with TBI before 24 h of hospital arrival and patients with pre-existing neurological disorders were excluded from the study.

### 2.3. Main Outcomes

The primary outcome measure was mortality at hospital discharge. The secondary outcomes were degrees of disability and quality of life, assessed at hospital discharge, measured using the modified Rankin Scale (mRs) [16,17]. Poor functional recovery was defined by mRs scores of 4 (moderately severe disability), 5 (severe disability), and 6 (death).

### 2.4. Variables and Measurements

The primary exposure of this study was whether fever at the time of injury, defined as a body temperature of 38 °C or higher, was present at the scene, or at the ED if the fever measured at the scene after brain injury was unknown. Body temperature was measured within 5 min after ED admission, and the patient with fever at the time of initial measurement was defined as the fever group. Blood and urine cultures were collected for the investigation of the cause of the fever, and the results of the blood culture were first reflected and classified into bacterial and viral diseases.

Patient characteristics were obtained from chart review by the emergency physician. Patient demographics included age (18–64 vs. 65–), sex, and comorbidities (e.g., hypertension, diabetes mellitus, etc.). Prehospital and hospital information surveyed included mechanism and place of injury; injury severity, defined as an abbreviated injury scale (AIS) and new injury severity score (NISS); mental status; type of brain hemorrhage; electrolyte levels; osmolality; and clinical outcomes on discharge from the ED. Additionally, an AIS score of TBI of 3 or more was defined as severe TBI.

### 2.5. Statistical Analysis

Patient demographics, injury demographics, prehospital treatment, hospital treatment, and clinical outcomes according to fever occurrence after TBI were compared using the Wilcoxon rank-sum test for continuous variables and the chi-squared test for categorical variables. Univariable and multivariable logistic regression analyses were performed to estimate the effect sizes of fever on mortality at hospital discharge and functional disability. The adjusted potential confounders identified in directed acyclic graph (DAG) models included age, gender, seasonality, comorbidities including hypertension and diabetes mellitus, mechanism of injury, place of injury, and severity of TBI.

Crude and adjusted odds ratios (ORs) with corresponding 95% confidence intervals (CIs) were calculated. Finally, interaction analysis was performed between fever occurrence and age group on study outcomes to find out whether the association between the incidence of fever and the study outcomes of TBI varies with age (fever occurrence × age group). All statistical analyses were performed using SAS version 9.4 (SAS Institute Inc., Cary, NC, USA).

## 3. Results

After excluding patients with unknown information regarding fever and/or study outcome, a total of 690 patients were enrolled in our registry for the final analysis (Figure 1).

The demographics of the study population according to the fever are shown in Table 1. Fever was observed in 18.70% (129/690) of the study population. The proportion of mortality at hospital discharge was 10.5% (14/129) in the fever group and 9.7% (54/561) in the non-fever group (*p*-value = 0.55).

The characteristics of the study population according to the age group are shown in Table 2. Fever was observed in 20.2% (88/438) of patients in the 18–64-years-old group (younger age group) and 16.2% (41/252) of patients in the >65-years-old group (elderly group). The proportion of mortality at hospital discharge was 9.3% (41/438) in the younger age group and 10.8% (27/252) in the elderly group.

### 3.1. Main Results

In the multivariable logistic regression analysis, compared with patients without fever, patients with fever had no significant difference in mortality at hospital discharge (aOR, 95% CIs: 1.24 (0.57–3.02)) and poor neurological recovery (0.84 (0.44–1.54)) after the full adjustment of potential confounders (Table 3).

### 3.2. Interaction Analysis

In interaction analysis, assessing whether study outcomes by fever vary according to age, the ORs for mortality at hospital discharge of the fever group differed from that of the non-fever group depending on the age of patients with TBI (*p* for interaction < 0.01) (Table 4). In younger patients, the incidence of fever had no significant effect on the mortality at discharge (aOR (95% CIs): 0.85 (0.51–1.54)), whereas the incidence of fever significantly increased mortality in the elderly group (1.39 (1.13–1.50)).

## 4. Discussion

We investigated the impact of fever on the clinical outcomes of patients with TBI and tested the interaction effect of fever on study outcomes according to age group. In our study, fever was not associated with mortality and poor functional recovery in patients with TBI. However, the interaction analysis showed that fever significantly increased mortality in the elderly group of patients with TBI.

Fever is a common condition in patients with brain injuries, observed in 20–50% of patients with TBI, and nearly 90% of patients have at least one episode within seven days of hospitalization [18].

In previous studies, fever was associated with worse outcomes regardless of the etiology of the brain injury. In ischemic stroke, fever on hospital admission is associated with higher mortality than normothermia and is independently associated with the severity of the stroke, infarction size, and clinical outcomes [19]. In a study of spontaneous brain hemorrhages such as subarachnoid hemorrhage and intracranial hemorrhage, fever contributed to worse survival and neurological outcomes [20,21]. In patients with TBI, fever in the early stage of injury is an independent prognostic predictor for TBI and is associated with lowering functional recovery, and the burden of fever affecting the prognosis was more significant than that of age [22]. The meta-analysis for neurologic injuries such as hemorrhagic stroke, ischemic stroke, and TBI indicated that fever was significantly associated with worse outcomes, as indicated by higher mortality rates, worse functional outcomes, greater severity, more significant disability, more dependence, and more extended stays in the hospital and ICU [7].

While infection is the most common cause of fever, fever in patients with TBI may have multiple causes, not only from infection; more importantly, fever may be due to the disruption of the hypothalamic set point by endogenous pyrogen released from injured neurons [7]. Furthermore, fever may affect the patient through several mechanisms, including electrolyte disturbance, the inhibition of protein kinases, cytoskeletal proteolysis, free radical production, excite-toxicity, and blood–brain barrier breakdown, leading to cerebral edema, potentially decreasing CPP [7,8]. Another possible mechanism is that fever increases the cerebral metabolic rate for oxygen and glucose, and this rate’s increase leads to an increase in cerebral blood flow in a compromised brain that has impaired autoregulation and hence to an increase in cerebral blood volume and intracerebral pressure [13].

Although the mechanism is not clear, aging is known to be a factor that worsens the clinical outcomes of patients with TBI. In one study that presented mortality according to age in patients with TBI, 30-day mortality in the age groups 15–54 years, 55–64 years, 65–74 years, 75–84 years, and ≥85 years was 6%, 11%, 11%, 23%, and 24%, respectively [23]. Elderly patients over 65 years of age showed poorer functional recovery and increased mortality at hospital discharge, as in previous studies.

In the interaction analysis for total patients with TBI, fever increased mortality only in elderly patients over 65 years of age. There are no studies on the effect of fever as a poor prognostic factor in elderly TBI patients. However, a possible explanation for the observed higher morbidity and mortality rates among elderly patients includes low physiological reserves due to the biological changes that accompany aging and the frequent presence of comorbid illnesses [24]. Another possible explanation is that cerebral edema formation in TBI increases faster and more severely with increasing age, as demonstrated in a mice study. In the study, the development of cerebral edema was accelerated when an elderly TBI mouse had a fever [11].

In our study, fever in elderly patients with TBI was associated with mortality at hospital discharge. In addition to GCS and brain CT, which are the traditional prognostic factors of TBI, the fever occurrence should be carefully observed, especially in elderly patients, since fever in elderly patients can be an early prognostic sign of poor prognosis. To reduce the burden of TBI, further studies are needed to determine the prognosis of patients with TBI rapidly and to develop strategies to improve mortality after injury, especially in elderly TBI populations.

This study has several limitations. First, previous studies suggested various criteria for fever, the primary factor of our study. In our study, fever was defined as a body temperature above 38 °C. Due to this setting, it is possible that the occurrence of fever was under- or overestimated. Second, other serum parameters related to infection, such as white blood cells or C-reactive protein, could not be investigated and there were many missing values for the cause of fever. Third, the body temperature measured immediately after ED arrival was defined as the body temperature at the time of brain injury, and changes in the body temperature during transport were not considered. Fourth, it was assumed that the clinical outcomes would differ depending on whether the fever was controlled or not. However, interaction analysis could not be performed owing to the small number of patients with fever control, and it was not possible to adjust because it was not shown to be a confounder on the directed acyclic graph. Fifth, though the ratio of antipyretic use for patients with fever was presented, there were no specific data on the type, time of use, or dosage of antipyretic drugs. The use of antipyretics may have influenced the results of this study. Sixth, the Glasgow Outcome Scale-Extended (GOSE) has become one of the most widely used outcome instruments to assess global disability and recovery after TBI. However, GOSE could not be collected in our registry and, therefore, there were limitations in presenting patients’ clinical outcomes with various indicators. Finally, as the study design was not a randomized controlled trial, there may be significant potential biases that were not controlled.

## 5. Conclusions

In our study, fever was associated with mortality in elderly patients with TBI. Although further investigation is needed on the causal association, rapid workup about fever origin and fever control is needed to reduce the burden of TBI.

## Figures and Tables

**Figure 1 medicina-58-01860-f001:**
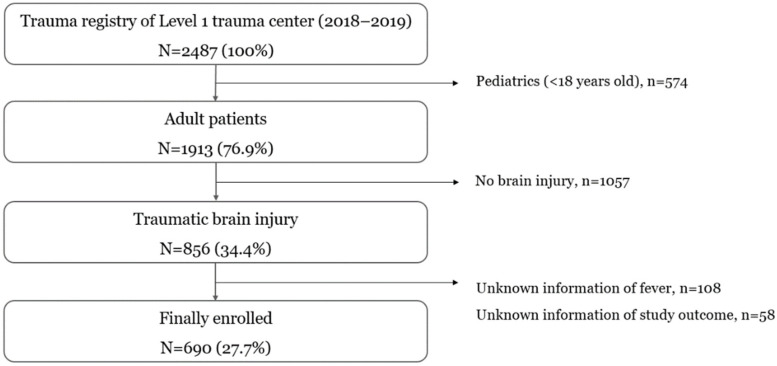
Study populations.

**Table 1 medicina-58-01860-t001:** Characteristics and clinical outcomes between traumatic brain injury patients with and without fever.

Variables	All	Fever > 38 °C		
		Yes	No	*p*-Value
	N (%)	N (%)	N (%)	
All	690 (100.0)	129 (100.0)	561 (100.0)	
Age				0.04
18–64	438 (63.5)	88 (68.4)	350 (62.4)	
65–	252 (36.5)	41 (31.6)	211 (37.6)	
Gender				<0.01
Female	218 (31.5)	60 (46.1)	158 (28.2)	
Season, summer	174 (25.2)	31 (24.0)	143 (25.5)	0.11
Underlying disease				
Hypertension	310 (44.9)	63 (48.7)	247 (43.9)	0.06
Diabetes	193 (28.0)	37 (28.9)	156 (27.9)	0.46
Mechanism of injury				0.25
Traffic	240 (34.7)	46 (35.5)	194 (34.5)	
Fall down	313 (45.3)	51 (39.5)	262 (46.7)	
Other	137 (20)	32 (25.0)	105 (18.8)	
Place of injury				0.02
Home	212 (30.7)	27 (21.1)	185 (33.0)	
Street	256 (37.1)	49 (38.1)	207 (37.0)	
Other	222 (32.2)	53 (40.8)	169 (30.0)	
Time from injury to hospital admission, hour median (IQR)	2 (1–4)	2 (1–3)	2 (1–4)	0.07
Severity (AIS ≥ 3)				
High AIS score of TBI	140 (20.3)	19 (14.5)	121 (21.5)	<0.01
High AIS score of other region	70 (10.1)	24 (18.4)	46 (8.2)	<0.01
Severity of trauma (NISS)				0.01
1–8	70 (10.1)	24 (18.4)	46 (8.2)	
9–15	148 (21.4)	17 (13.2)	131 (23.3)	
16–24	199 (28.9)	46 (35.5)	153 (27.3)	
25–75	273 (39.6)	43 (32.9)	230 (41.2)	
Glasgow coma scale				0.1
15, alert	377 (54.6)	69 (53.5)	308 (54.9)	
13–14, drowsy	135 (19.6)	25 (19.4)	110 (21.0)	
8–12, stupor	104 (15.1)	19 (14.7)	85 (15.2)	
3–7, coma	74 (10.7)	16 (12.4)	58 (8.9)	
q-SOFA, (≥2)	158 (22.9)	35 (27.1)	123 (21.9)	0.04
Sodium (mmol/L)				0.09
<135	95 (13.8)	19 (14.5)	77 (13.6)	
135–145	595 (86.2)	111 (85.5)	485 (86.4)	
Potassium (mmol/L)				<0.01
–3.5	148 (21.4)	24 (18.4)	124 (22.1)	
3.5–5.0	517 (74.9)	99 (76.3)	418 (74.5)	
5.0–	25 (3.7)	7 (5.3)	18 (3.4)	
Osmolality (mmol/kg)				<0.01
–275	61 (8.9)	8 (6.6)	53 (9.4)	
275–295	401 (58.1)	85 (65.8)	316 (56.4)	
295–	228 (33.0)	36 (27.6)	192 (34.2)	
Prehospital management				
Advanced airway	58 (8.4)	11 (8.5)	47 (8.4)	0.38
Fluid resuscitation	71 (10.3)	17 (13.2)	54 (9.7)	0.05
ED management				
Surgical intervention	137 (19.9)	27 (20.9)	110 (19.6)	0.14
Blood transfusion	77 (11.2)	15 (11.6)	62 (11.0)	0.54
IV drugs for fever control	77 (11.2)	44 (34.1)	33 (5.9)	<0.01
IV drugs for hypertension	148 (21.4)	38 (29.4)	110 (19.6)	0.03
IV drug for hypotension	66 (9.6)	14 (10.8)	52 (9.3)	0.27
Origin of fever (blood or urine culture)				<0.01
Bacterial disease	152 (22.1)	78 (60.4)	74 (13.2)	
Viral disease	90 (13.0)	38 (29.5)	52 (9.3)	
Other or unknown	448 (64.9)	13 (10.1)	435 (77.5)	
Outcomes				
Poor functional recovery	211 (30.5)	34 (26.3)	177 (31.5)	0.02
Mortality at hospital discharge	68 (9.9)	14 (10.5)	54 (9.7)	0.55

AIS, abbreviated injury scale; TBI, traumatic brain injury; NISS, new injury severity score; ED, emergency department; IV, intravenous; q-sofa, quick sepsis-related organ failure assessment.

**Table 2 medicina-58-01860-t002:** Characteristics and clinical outcomes between traumatic brain injury patients according to age.

Variables	All	Age		
		18–64	65-	*p*-Value
	N (%)	N (%)	N (%)	
All	690 (100.0)	438 (100.0)	252 (100.0)	
Fever > 38 °C				0.04
Yes	129 (18.7)	88 (20.2)	41 (16.2)	
Gender				0.34
Female	218 (31.5)	123 (28.1)	95 (37.8)	
Season, summer	174 (25.2)	105 (24.0)	69 (27.4)	0.11
Underlying disease				
Hypertension	310 (44.9)	136 (31.1)	173 (68.9)	0.06
Diabetes	193 (28.0)	95 (21.7)	98 (39.0)	0.46
Mechanism of injury				0.25
Traffic	240 (34.7)	165 (37.6)	75 (29.8)	
Fall down	313 (45.3)	194 (44.2)	119 (47.4)	
Other	137 (20.0)	80 (18.2)	57 (22.8)	
Place of injury				0.2
Home	212 (30.7)	110 (25.1)	102 (40.5)	
Street	256 (37.1)	172 (39.2)	84 (33.4)	
Other	222 (32.2)	156 (35.7)	66 (26.1)	
Time from injury to hospital admission, hour median (IQR)	2 (1–4)	2 (1–4)	2 (1–4)	0.43
Severity (AIS ≥ 3)				
High AIS score of TBI	140 (20.3)	92 (21.0)	48 (18.9)	0.11
High AIS score of other region	70 (10.1)	43 (9.7)	27 (10.8)	<0.01
Severity of trauma (NISS)				0.01
1–8	70 (10.1)	43 (9.7)	27 (10.8)	
9–15	148 (21.4)	85 (19.4)	63 (25.0)	
16–24	199 (28.9)	143 (32.6)	56 (22.3)	
25–75	273 (39.6)	167 (38.3)	106 (41.9)	
Glasgow coma scale				<0.01
15, alert	377 (54.6)	254 (58.0)	123 (48.8)	
13–14, drowsy	135 (19.6)	91 (20.8)	44 (17.5)	
8–12, stupor	104 (15.1)	62 (14.2)	42 (16.7)	
3–7, coma	74 (10.7)	31 (7.1)	43 (17.1)	
q-SOFA, (≥2)	158 (22.9)	100 (22.8)	58 (23.0)	0.48
Sodium (mmol/L)				0.09
<135	95 (13.8)	36 (8.2)	59 (23.4)	
135–145	595 (86.2)	403 (92.0)	192 (76.4)	
Potassium (mmol/L)				<0.01
–3.5	148 (21.4)	94 (21.3)	54 (21.6)	
3.5–5.0	517 (74.9)	333 (76.1)	184 (73.0)	
5.0–	25 (3.7)	12 (2.7)	14 (5.6)	
Osmolality (mmol/kg)				<0.01
–275	61 (8.9)	8 (6.6)	32 (12.8)	
275–295	401 (58.1)	264 (60.2)	138 (54.7)	
295–	228 (33.0)	146 (33.4)	82 (32.4)	
Prehospital management				0.38
Advanced airway	58 (8.4)	35 (8.0)	23 (9.1)	
Fluid resuscitation	71 (10.3)	43 (9.7)	29 (11.5)	0.35
ED management				
Surgical intervention	137 (19.9)	96 (21.9)	41 (16.3)	0.04
Blood transfusion	77 (11.2)	57 (13.0)	20 (7.9)	<0.01
IV drugs for fever control	77 (11.2)	44 (10.0)	33 (13.1)	
IV drugs for hypertension	148 (21.4)	65 (14.8)	83 (33.0)	
IV drug for hypotension	66 (9.6)	34 (7.8)	32 (12.7)	
Origin of fever (blood or urine culture)				0.17
Bacterial disease	152 (22.1)	96 (21.9)	56 (22.2)	
Viral disease	90 (13.0)	57 (13.0)	33 (13.1)	
Other or unknown	448 (64.9)	285 (65.1)	163 (64.7)	
Outcomes				
Poor functional recovery	211 (30.5)	128 (29.1)	83 (33.1)	
Mortality at hospital discharge	68 (9.9)	41 (9.3)	27 (10.8)	0.55

AIS, abbreviated injury scale; TBI, traumatic brain injury; NISS, new injury severity score; ED, emergency department; IV, intravenous; q-sofa, quick sepsis-related organ failure assessment.

**Table 3 medicina-58-01860-t003:** Multivariable logistic regression model for study outcomes.

Study Outcomes		Total	Outcome		Model 1	Model 2	Model 3
		N	N	%	aOR (95% CI)	aOR (95% CI)	aOR (95% CI)
Poor neurological recovery							
Fever > 38 °C	No	561	177	31.5	1.00	1.00	1.00
	Yes	129	34	26.3	0.88 (0.50–1.56)	0.90 (0.50–1.60)	0.84 (0.44–1.54)
Age	18–64	438	128	29.1	1.00	1.00	1.00
	65-	252	83	33.1	1.57 (0.96–2.55)	1.65 (1.01–2.71)	1.67 (1.11–2.76)
Mortality at hospital discharge							
Fever > 38 °C	No	561	54	9.7	1.00	1.00	1.00
	Yes	129	14	10.5	1.37 (0.58–3.19)	1.30 (0.55–3.08)	1.24 (0.57–3.02)
Age	18–64	438	41	9.3	1.00	1.00	1.00
	65-	252	27	10.8	1.56 (1.15–2.01)	1.49 (1.19–3.02)	1.50 (1.20–1.75)

aOR, adjusted odds ratio; CI, confidence interval. Model 1: adjusted age, gender, hypertension, and diabetes mellitus. Model 2: adjusted variables of model 1 + mechanism of injury, and place of injury. Model 3: adjusted variables of model 2 + advanced airway, fluid resuscitation, seasonality, and severity of TBI.

**Table 4 medicina-58-01860-t004:** Interaction analysis between fever and age group.

	Fever > 38 °C				
	No	Yes			*p*-for Interaction
Poor neurological recovery		aOR	95% CI		
Age					0.60
18–64	ref.	0.68	0.32	1.41	
65–	ref.	1.39	0.52	3.70	
Mortality at discharge					
Age					<0.01
18–64	ref.	0.85	0.51	1.54	
65–	ref.	1.39	1.13	1.50	

aOR, adjusted odds ratio; CI, confidence interval. Adjusted for age, gender, seasonality, hypertension, diabetes mellitus, mechanism of injury, place of injury, advanced airway, fluid resuscitation, and severity of TBI.

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
