# Peer review of "Effect of Fever on the Clinical Outcomes of Traumatic Brain Injury by Age"

_medicina, 2022, doi:10.3390/medicina58121860_

Round 1

Reviewer 1 Report (Previous Reviewer 2)

GCS is essential for a TBI study at the time of admission , also the time taken from injury to admission. 

SOFA scores and cultures are missing although mentioned in limitations.

Author Response

(1) GCS is essential for a TBI study at the time of admission, also the time taken from injury to admission. 

ANSWER: Thank you for the review. We added variables of GCS and time from injury to hospital admission in Table 1 and 2 according to your advice.

(REVISION: Table 1 and 2)

(2) SOFA scores and cultures are missing although mentioned in limitations.

ANSWER: Thank you for the review. Variables for calculating SOFA score such as FiO2 and bilirubin were not collected in our registry, so q-SOFA score, which can be calculated, was added to the variables.

The results of blood and urine culture had many unknown values, so they were excluded from the original manuscript, but added to the revised manuscript as you advised.

(REVISION: Table 1 and 2)

(REVISION: Materials and Methods-Variables and measurements): Blood and urine culture were collected for the investigation of cause of fever, and the results of blood culture were first reflected and classified into bacterial and viral diseases.

(REVISION: Discussion): Second, other serum parameters related to infection such as white blood cells or C-reactive protein, could not be investigated and there were many missing values for the cause of fever.

Reviewer 2 Report (Previous Reviewer 1)

The authors did a good job revising the manuscript and the manuscript has been significantly improved. 

After following minor revision, this manuscript warrants publication in Medicina.

1. Line 175: Consider using a more appropriate transition word (And, interaction analysis showed that fever significantly increased mortality in the elderly group of patients with TBI.) => e.g., However,...

2. Lines 126-128:

The sentence needs to be revised. Suggestion: After excluding patients with unknown information of fever and/or study outcome, a total of 690 patients were enrolled in our registry for the final analysis (Figure 1). 

Author Response

The authors did a good job revising the manuscript and the manuscript has been significantly improved. After following minor revision, this manuscript warrants publication in Medicina.

  1. Line 175: Consider using a more appropriate transition word (And, interaction analysis showed that fever significantly increased mortality in the elderly group of patients with TBI.) => e.g., However,...

ANSWER: Thank you for the review. We revised sentence according to your advice.

(REVISION: Discussion): However, interaction analysis showed that fever significantly increased mortality in the elderly group of patients with TBI.

  1. Lines 126-128: The sentence needs to be revised. Suggestion:After excluding patients with unknown information of fever and/or study outcome, a total of 690 patients were enrolled in our registry for the final analysis (Figure 1). 

ANSWER: Thank you for the review. We revised sentence according to your advice.

(REVISION: Results): After excluding patients with unknown information of fever and/or study outcome, a total of 690 patients were enrolled in our registry for the final analysis (Figure 1).

This manuscript is a resubmission of an earlier submission. The following is a list of the peer review reports and author responses from that submission.

Round 1

Reviewer 1 Report

While there are many studies on the effect of fever on TBI, there is limited evidence on the effect of fever on clinical outcomes of TBI according to different age groups. This study investigated the significant interaction effect between age and fever on the clinical outcomes of TBI. Results of this manuscript has merits and will be of interest to many readers. However, this manuscript may be improved with several minor revisions.

Abstract:

- Throughout the paper, Emergency medical service (EMS) -> emergency medical services

- Line 13-14 "The main exposure is fever, defined as body temperature of 38C should say "The main exposure was fever, defined as body temperature of 30C or above"

- Lines 19 & 20: when mentioning elderly patients and younger patients, should indicate actual age group. e.g., elderly patients (>65 years), younger patients (18-64 years)

- Lines 22-23: Authors concluded that "Therefore, rapid workup regarding origin of fever control is needed to reduce the burden of TBI", but this conclusion statement does not really reflect the main results of the study. 

Introduction

- Lines 28, 30, etc.: Throughout the manuscript, authors need to double check for correct citation format. For citation [1] and [2], a space is needed before the citation. Lines 182, 195, 201, 204 also need correct citation format. 

- Throughout the manuscript, English needs to be polished. For example, Lines 31-34 are incomplete sentence and needs to be revised. Lines 36-39 are also incorrect sentence structure. 

- Lines 59-60, "This effect differs across..." Is this your hypothesis? The sentence needs to be revised.

Materials and Methods 

- Study design, setting, and data sources: Study period (time period for data collection) needs to be indicated. (e.g., data from when to when?)

- Variables and measurements: It says that the primary exposure of this study  is "fever at the time of injury", which was defined as body temperature of 38C or higher on arrival at the ED after brain injury". What is average time from the brain injury to ED arrival? Is it reasonable to say that the body temperature/fever measured on ED arrival is proxy for body temperature/fever at the time of injury?

- Data on various variables, including patient demographics, prehospital and hospital information, were collected and adjusted in the model. In addition, have you examined whether season/month of injury is also confounding factor for the association between fever and TBI outcome as environmental temperature may be associated with both body temperature and clinical outcomes?

Results

- Lines 129-131 says "All were included because there was no unknown information of fever and study outcome variables (Figure 1)." However, this doesn't match with Figure 1, which shows there were unknown information of fever (n=108) and unknown information of study outcome (n=58).

- Results don't contain subgroup analysis based on the severity of TBI (non-severe TBI vs. severe TBI patients). Throughout Discussion section, authors state that in this study, fever increased mortality and reduced functional recovery only in patients with severe TBI. However, there is no indication of having subgroup-analysis based on severity of TBI in neither methods nor results section. 

- Table 3&4: Did you consider adjusting for pre-hospital management factor into any of the models? Also, Table 4 needs to have a footnote describing /listing adjusting factors for aOR.  

Discussion

-As mentioned earlier, the claims on observing effect of fever on clinical outcomes of TBI patients only in patients with severe TBI is not supported by results as authors did not present any results on this. 

Reviewer 2 Report

Study is performed well.

Addition of WBC counts incl other hemogram parameters, CBC (blood, urine, other fluids as applicable) and culture reports is required esp. in conditions of fever.

Any organ failures, SOFA scores can add more to the study.

Whether surgical intervention was done, whether patient was on ventilator , blood transfusion etc would add more to interpretation.